# The Effect of the Open Vase-like Microcapsules Formation with NiFe Double-Hydroxide Walls during Hydrolysis of the Mixture NiSO$_4$ and FeSO$_4$ Salt Solution Microdroplets Deposited on the Alkaline Solution Surface

Valeri P. Tolstoy *, Alexandra A. Meleshko, Anastasia A. Golubeva and Elizaveta V. Bathischeva

Institute of Chemistry, Saint Petersburg State University, 198505 Saint Petersburg, Russia;
a.meleshko@spbu.ru (A.A.M.); st017485@student.spbu.ru (A.A.G.); st086578@student.spbu.ru (E.V.B.)
* Correspondence: v.tolstoy@spbu.ru; Tel.: +7-812-428-41-04

**Abstract:** In this work, the conditions for the synthesis of open vase-like microcapsules with a size of 1–5 μm and 20–40 nm walls of NiFe$_{0.3}$(OH)$_x$ layered double hydroxide were studied. These microcapsules were obtained by the rapid hydrolysis of microdroplets of a solution of a mixture of NiSO$_4$ and FeSO$_4$ salts at the surface of an alkali solution. A hypothetical model of successive chemical processes occurring at the interface during synthesis is presented. The features of the "rim" formation around each microcapsule hole from the wall material with a peculiar nozzle-like shape are noted. These microcapsules can be transferred to the surface of a nickel foil using the Langmuir–Schaefer (LS) method. During the transfer process, they are fixed to the surface in an oriented position with a "rim" that contacts the nickel surface. It was established that electrodes made of such a foil with a layer of microcapsules exhibit active electrocatalytic properties in the oxygen evolution reaction during the electrolysis of water in an alkaline medium.

**Keywords:** air–solution interface; aerosol; nickel and iron sulfates solution; vase-like microcapsules; NiFeOH layered double hydroxide; electrode for OER

## 1. Introduction

Great attention has been paid recently to the study of adsorption processes at the interface between aqueous solutions of metal salts or organic substances and gases, including air [1–8]. This is because of interest in such processes both from the side of classical physical and colloidal chemistry and from the preparative chemistry point of view involved in the synthesis of new compounds used in various nanomaterial compositions with unique properties. It has been noted in several articles, for example, in [9–12], that insoluble compound layers with a unique morphology and thickness gradients of density and composition are formed when a gaseous reagent interacts with a planar surface of several transition metal salts solutions. Then, because of the drying process, such layers can change their planar geometry and transform into tubular structures with a new set of unique properties. The results of [13] are also noteworthy, because it was found that when the surface of a Mn(OAc)$_2$ solution drop is treated with ozone, ordered honeycomb-like nanosheets of manganese (IV) oxide are formed with a birnessite structure. There are also known experiments on the synthesis of hollow microspheres with walls of calcium phosphate or copper hydroxide with a size of several microns formed by the interaction of microdroplets of calcium or copper salts with the surface of aqueous solutions of ammonium phosphate or alkali, as well as the synthesis of hollow microspheres of zirconium oxide by the interaction of ammonium hydroxide microdroplets with the surface of an alcohol solution of zirconium alkoxide [14]. As noted in the abovementioned article, as a result of several hours of interaction at the interface and in the volume of the solution, precipitates of the corresponding compounds

are formed at the bottom of the container, which partially contain hollow microspheres in composition. The creation of microcapsules with such morphology and walls from a wide range of inorganic compounds is an important issue of preparative chemistry, since such materials have unique properties and can form the basis of many electrochemical devices for energy and sensors, as well as exhibit the properties of active catalysts and sorbents.

Previously, we showed [15] that when a solution of a mixture of $NiSO_4$ and $FeSO_4$ salts was sputtered as an aerosol onto the surface of an alkali solution, open microcapsules with walls of layered double hydroxide with the general formula $Ni_xFe(OH)_y$ and the unique vase-like morphology are formed.

The main aim of this work was to optimize the conditions for such synthesis and obtain ordered arrays of such microcapsules on the nickel electrode surface.

## 2. Materials and Methods

### 2.1. Materials, Reagents, and Synthesis Conditions

Aqueous solutions of $NiSO_4·7H_2O$, $FeSO_4·7H_2O$, NaOH (provided by Vekton) and $NaBH_4$ (Aldrich) were used as reagents for synthesis. Solutions were prepared via the dissolution of weighed portions of chemicals with at least 30 min stirring. Two solutions were used for microcapsule synthesis. The first one was a mixture of 0.4 M solution of $NiSO_4$ with 0.1 M solution of $FeSO_4$, which was sputtered as an aerosol, and the second one was a mixture of 1 M solution of NaOH with 0.5 M solution of $NaBH_4$, which was put in a planar Teflon vessel. This ratio of the concentrations of Ni(II) and Fe(II) salts was chosen in accordance with the recommendations of the works [16,17]. It is shown that the best electrocatalytic properties are achieved for close values of the ratio of the concentrations of these elements in the electrocatalyst layer. Sputtering was carried out using an A&DUN-231 ultrasonic nebulizer with a generator with a power of 10 W and a frequency of 2.5 MHz. The sputtering chamber and the Teflon vessel were interconnected via a L-shaped glass tube with an inner diameter of 15 mm and a length of up to 15 cm, with a conical flare over the Teflon vessel (Figure 1). Duration of the treatment with the aerosol varied between 2 and 20 min. When the treatment was complete, the formed microcapsules were transferred from the solution surface to a nickel foil using the Langmuir–Schaefer method. A nickel foil sheet 0.5 mm thick had a mirror-like surface. The dimensions of the sample in the zone of contact with the alkali solution were 10 mm × 15 mm. The vertical movement of the foil was carried out using a special homemade semi-automatic manipulator at a speed of 0.5 mm/s. Then the resulting samples were washed with distilled water and ethyl alcohol and dried in air at 60 °C.

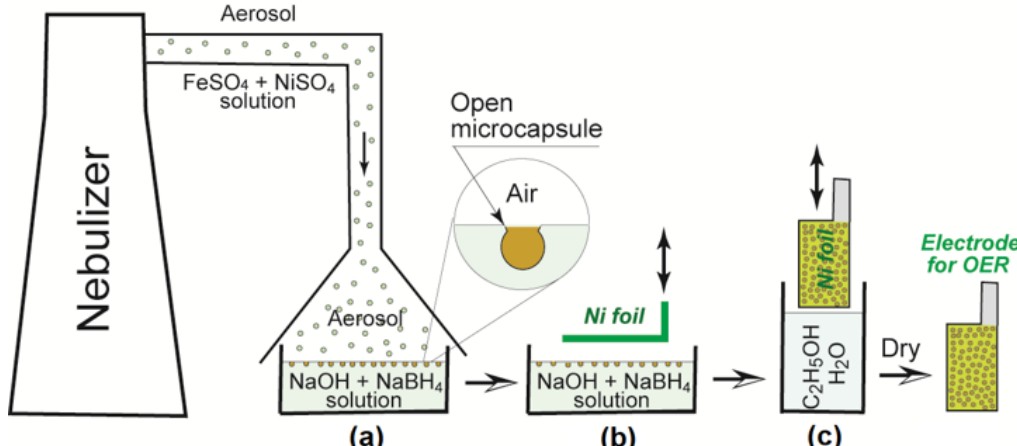

**Figure 1.** Schematic representation of the operations sequence in the preparation of microcapsules with walls of Ni(II) and Fe(III) double hydroxide: (**a**)—the stage of surface treatment of a NaOH and $NaBH_4$ mixture solution with microdroplets of an aerosol from a mixture solution of $NiSO_4$ and $FeSO_4$; (**b**)—transferring stage of a microcapsules array to the nickel foil surface according to the LS method; (**c**)—the stage of sample washing with distilled water and ethyl alcohol.

Several samples of electrocatalysts were also prepared using the drop-casting method. The solution of a mixture of $NiSO_4$ and $FeSO_4$ salts was added drop by drop to the solution of a mixture of NaOH and $NaBH_4$ and the formation of a precipitate was observed. Then, the precipitate was separated from the initial solution and washed three times with water. Next, ethyl alcohol was added to the wet precipitate and a suspension was obtained, which was applied drop by drop to the surface of the nickel foil. After drying in air at 60 °C, these samples were ready for electrochemical studies.

### 2.2. Physical Characterization

Electron micrographs were obtained using a Zeiss Auriga Laser microscope at an accelerating voltage of 4 keV and a Zeiss Libra 200 transmission electron microscope. Microcapsule walls were etched using the FIB system of a Zeiss Auriga Laser microscope in a $Ga^+$ ion beam with an energy of 30 keV with the simultaneous process monitoring using the SEM method. The composition of the microcapsule walls was determined using the XPS method on an ESCALAB 250Xi electron spectrometer with Al K$\alpha$ radiation (14.866 eV) and by EDX microanalysis using an Oxford Instruments X-Max 80 microprobe, which is included in the transmission electron microscope kit.

### 2.3. Electrochemical Measurements

The electrocatalytic properties of the obtained electrodes composed of applied microcapsules on the nickel plate surface were studied via cyclic voltammetry using linear potential scanning for oxygen evolution reaction (OER) during the water electrolysis. The characteristics of these electrodes were measured using an Elins P-45X-FRA-24M potentiometer and a three-electrode cell. The nickel foil with a layer of microcapsules was used as the working electrode, whereas a graphite rod and a Hg/HgO electrode acted as a counter and the reference electrodes, respectively. The measurements were performed in 1 M aqueous solution of KOH at room temperature and atmospheric pressure, with a scanning rate of 10 mV/s. The Nernst equation was used to calculate the overpotential value. Electrochemical measurements were performed with iR compensation.

### 3. Results and Discussion

As follows from the micrographs shown in Figure 2a,d–f, the hydrolysis of the mixture of $NiSO_4$ and $FeSO_4$ in the microdroplets sputtered at the surface of the alkali solution resulted in the formation of hollow Vase-Like Microcapsules (VLM) with sizes of 1–5 μm. The density of their location at the interface depends on the time of the aerosol sputtering. Thus, experience shows that separate VLMs are formed at the interface in a few minutes, microcapsules with a density of about $10^5$ pcs/mm$^2$ are formed in 8–10 min, and at a time of more than 10–12 min a continuous layer of the synthesized substance is formed, which contains individual fragments of the microcapsules. Apparently, in the VLM walls formation process over 10–12 min or more, the effects of the repeated entry of aerosol drops into each of the areas of the alkaline solution surface are observed, and due to this, the drops form a continuous layer of nickel and iron oxyhydroxide at the interface. With a processing time of 8–10 min or less, the probability of multiple drops from an aerosol falling into one area of the alkaline solution surface is less, and that is why an array of individual VLMs is formed on the surface of the solution. This circumstance determines the optimal time of treatment with an aerosol for 8 min.

In Figure 2b,c, it can be seen that the outer surface of the VLM consisted predominantly of spherical nanoparticles with size of 10–20 nm, whereas the inner surface contained the same nanoparticles along with the planer ones, with the lateral size of up to 50 nm. The investigation of the microcapsule walls by means of transmission electron microscopy (Figure 3a,b) actually confirmed the noted nanoparticles sizes. Moreover, the analysis of the high-resolution micrographs revealed that the nanopar-

ticles were crystalline (not shown in this figure). Unfortunately, it was not possible to obtain direct information on the interplanar distances in these experiments due to the so-called moiré pattern effect. At the same time, the electron diffraction study revealed that the nanoparticles were characterized by the interplanar distances of 0.24, 0.20, 0.14, and 0.10 nm (Figure 3c). According to [18], those values are close to the interplanar distances in β-Ni(OH)$_2$ with a trigonal crystalline structure and the $\overline{3}$m point groups.

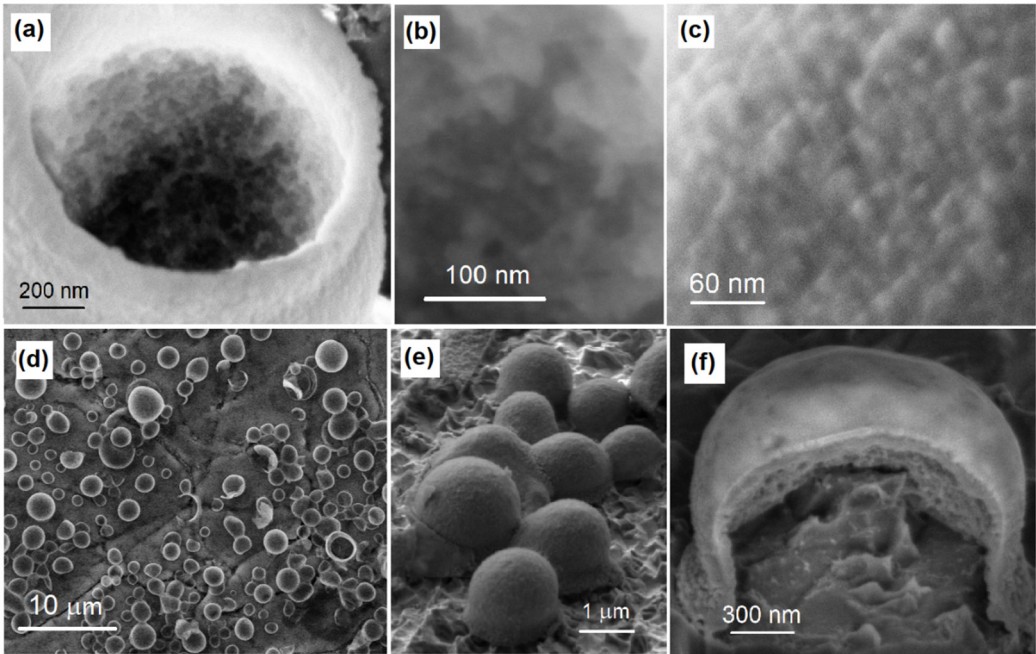

**Figure 2.** FESEM microphotographs of the microcapsules: (**a**)—general view of the hole in the microcapsule wall, (**b**)—inner surface of the microcapsule wall, (**c**)—outer surface of the microcapsule wall, (**d**)—top view of the microcapsules array on the electrode surface, (**e**)—top view at an angle of 45° on microcapsules array on the electrode surface, (**f**)—top view at an angle of 45° on the microcapsule on the electrode surface with the wall partially removed by etching with a Ga$^+$ ion beam.

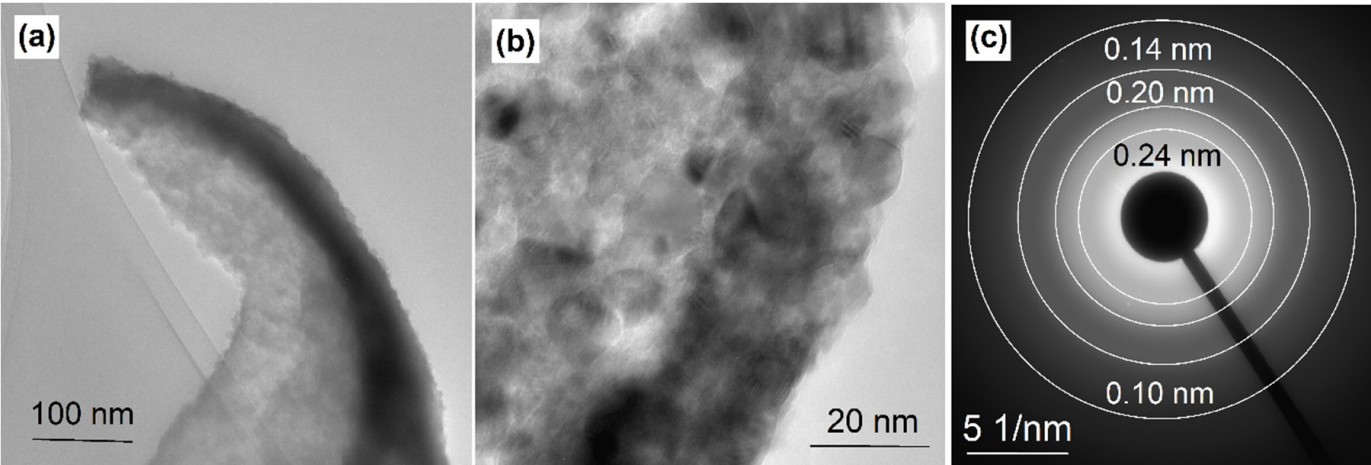

**Figure 3.** TEM (**a**,**b**) and SAED (**c**) images of nanoparticles forming the microcapsule wall.

In the EDX spectrum, the characteristic peaks of the electrons of Cu, C, Ni, Fe, and O atoms can be distinguished. The presence of the peaks of Cu and C atoms can be explained by the signal from the standard grid for TEM measurements, on which the test sample was applied. The ratio of the concentrations of the nickel and iron atoms in the walls was determined by EDX microanalysis (Figure 4a), and it turned out to be 3.4. It is noteworthy that this value is close to their ratio in the starting solution. According to XPS study, the composition of this layer includes Ni, Fe, and O atoms, with the position of the Fe$2p_{3/2}$ electron peak being 712.1 eV. That fact demonstrates the degree of oxidation of iron cations 3+. The position of the maximum of the Ni$2p_{3/2}$ electron peak at 855.4 eV indicates that its oxidation state is 2+ [19,20]. Apparently, the Fe(II) cations are oxidized during synthesis or upon the obtained sample exposure to air, while the Ni(II) cations do not change their oxidation state under these conditions. Attention is drawn to the asymmetric peak of 1$s$ oxygen electrons with maxima at 532.4 and 531.6 eV, which indicates the presence of oxygen in the layer in the composition of water molecules and M-OH (M = Fe, Ni) groups [20]. Thus, a comparison of the results obtained indicates the formation of microcapsule walls from nanocrystals of layered double hydroxide $NiFe_{0.3}(OH)_x$. It is important that the Na, S, and B atoms amount, which could be incorporated in the wall composition, does not exceed 5%. It confirms efficient side products separation by washing.

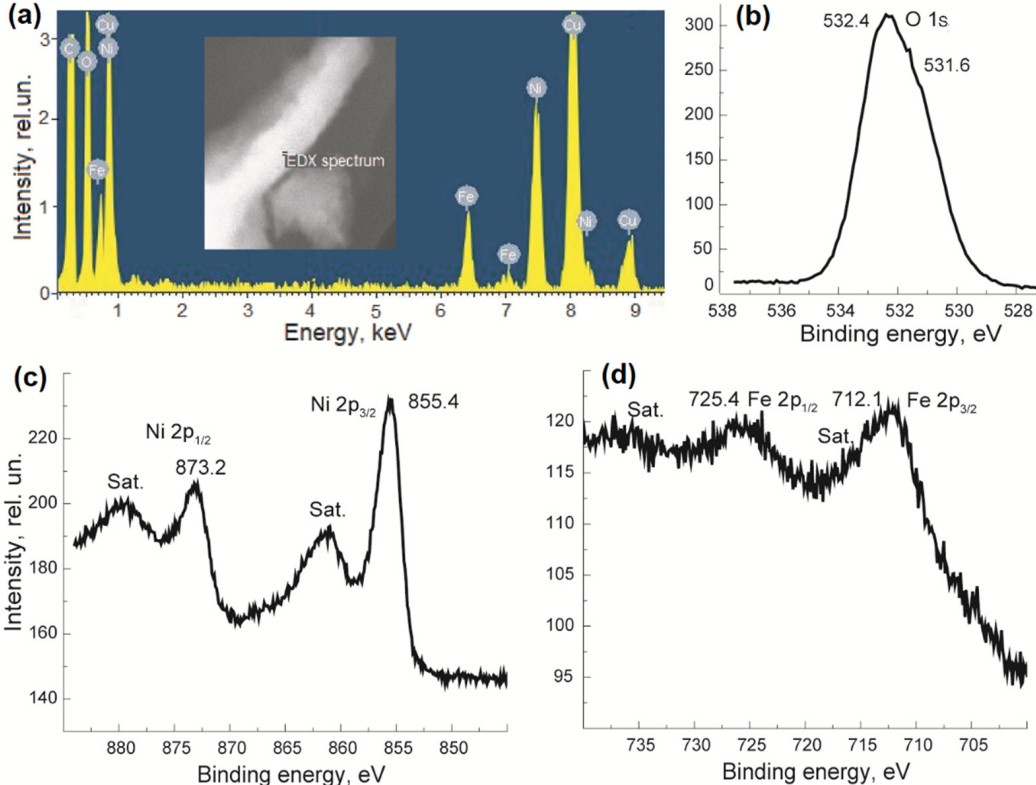

**Figure 4.** EDX spectrum (**a**) and XPS O1$s$ (**b**), Ni2$p$ (**c**), and Fe2$p$ (**d**) core levels spectra of nanoparticles forming the microcapsule wall.

Based on the experimental data obtained, it can be stated that when a microdrop of the solution of the $NiSO_4$ and $FeSO_4$ mixture reaches the surface of the 1 M alkali solution, fast hydrolysis of the salts occurs with the formation of the double nickel–iron hydroxide nanoparticles at the interface. Those nanoparticles form a relatively mechanically strong framework and, thus, the walls of the microcapsule are formed. At that stage, a peculiar semispherical "microcontainer" appears, its walls consisting of nanoparticles and immersed in the alkali solution. That "microcontainer" is open in the

upper part, since initially there is no contact between the nickel and iron cations with alkali in the upper part of the microdroplet, and the hydrolysis of the cations in that part is not observed. Due to this, open microcapsules are formed. Furthermore, the relatively fast formation of the porous microcapsule wall is followed by the ion diffusion and levelling of concentrations inside and outside the capsule, accompanied by relatively slow growth of the nanocrystals of the double hydroxide at the inner side of the capsule wall. Presumably, the nanocrystals of the double hydroxide with a planar morphology and a peculiar nozzle-like rim in the upper part of the microcapsule around its hole are formed on the inner surface at that stage (Figure 1a). On the one hand, the effect of such rim formation can be explained by the rapid hydrolysis of the nickel and iron cations and the formation of their sparingly soluble hydroxides in the region of the wave formation that arises around a drop when it enters a liquid solution [21]. On the other hand, this can be explained as a result of chemical reactions of the noted metal hydroxide formation on the meniscus surface around the microcapsule partially immersed in the alkali solution (Figure 1a).

The microbubbles of hydrogen can also be formed at the outer wall of such microcapsules. They are released on the surface of the nickel—iron hydroxide nanoparticles as a result of the $BH_4^-$ anions hydrolysis, and those microbubbles possibly prevented the microcapsules from the submersion in the alkali solution. It should be particularly noted that the size of the obtained microcapsules practically corresponds to the size of the microdroplets in the aerosol generated by the applied device, which confirms this sequence model of the chemical processes during the microcapsule formation.

Transfer of the microcapsules to the nickel foil using the LS method afforded an array of the VLMs which specified unique 3D texture of its surface. Moreover, as can be seen from the electron micrographs (Figure 2d–f), the VLMs on the electrode surface are located oriented with a hole towards the electrode. This becomes clear when taking into account the predominant orientation of the VLM at the solution–air interface with the hole towards the air and the horizontal position of the electrode plane when the VLM is transferred to it using the LS method.

Investigation of the electrocatalytic properties of the nickel electrodes obtained in this way showed that such electrodes exhibit active electrocatalytic properties towards the oxygen evolution reaction during water electrolysis in an alkaline medium (Figure 5). For example, for one of the samples, the overpotential value equaled 259 mV and the Tafel slope value was 64.8 mV/dec. It is important that values are significantly lower than the values for the original nickel sample (Figure 5). That overpotential value was only a few tens of millivolts higher than the values obtained for the similar electrocatalysts deposited on more expensive foamed nickel sample with developed interior surface and specially directed 3D morphology of the support [20]. It is substantially that the overpotential value of 259 mV is less than the similar value of 280 mV of the drop-casting sample. Apparently, this effect is observed due to the peculiarities of the morphology of the microcapsules, which contributes to the easier separation of oxygen bubbles from the electrode surface. According to other terminology, it provides relatively better aerophobic properties of the electrode.

It should be noted that the VLM electrodes have high stability of electrocatalytic properties. The study of the stability of electrocatalytic properties was carried out using the method of multiple cycling of the potential at a rate of 10 mV/s in the range of 0–600 mV vs. Hg/HgO electrode. It was shown that the overpotential value increased only by 25 mV after 100 such cycles.

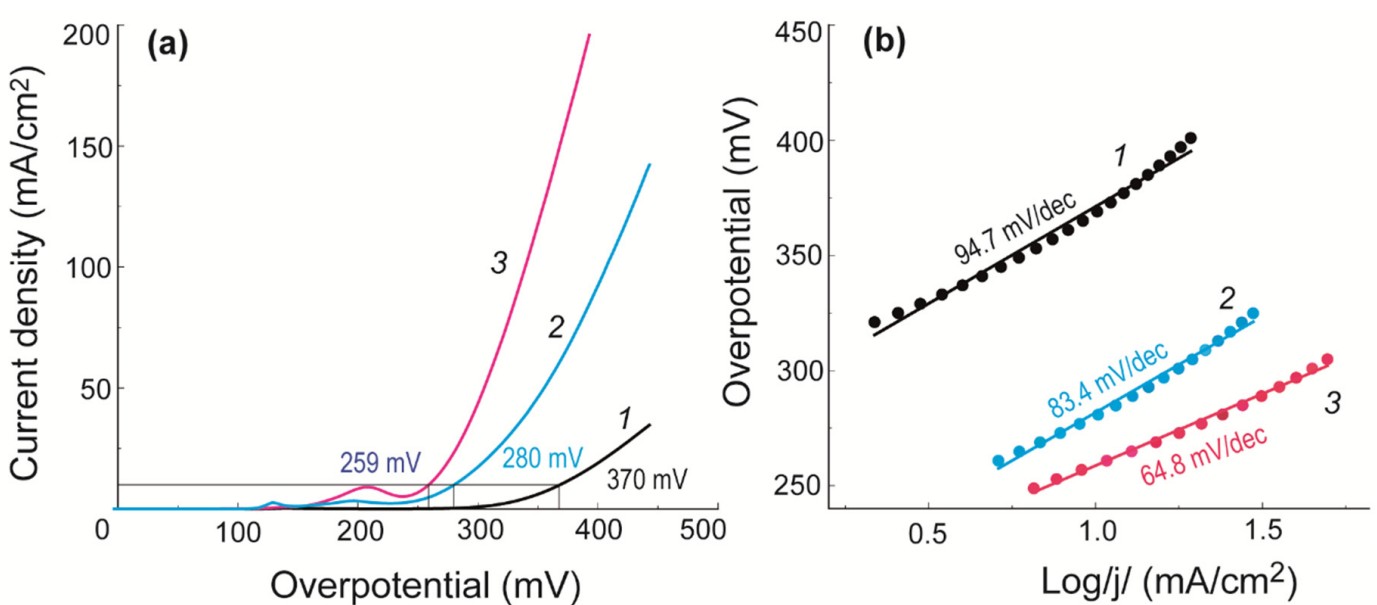

**Figure 5.** OER polarization curves (**a**) and Tafel plots (**b**) of the nickel electrodes. *1*—nickel foil, *2*—the layer of $NiFe_{0.3}(OH)_x$ nanoparticles on the nickel foil surface obtained by the drop-casting method, *3*—$NiFe_{0.3}(OH)_x$ VLM on the surface of nickel foil.

## 4. Conclusions

The open microcapsules with a unique vase-like morphology and a "rim" around the hole and with walls consisting of $NiFe_{0.3}(OH)_x$ LDH nanocrystals can be obtained by the rapid hydrolysis of microdroplets of the solution of the mixture of $NiSO_4$ and $FeSO_4$ salts at the surface of the alkali solution. These microcapsules at the alkali solution–air interface are oriented towards the direction of the open part towards the air and can be transferred to the surface of the nickel foil using the LS method. Moreover, obtained microcapsules are fixed on the substrate with orientation in the direction of the capsule open part to the substrate surface and form a kind of array on it. Nickel foil electrodes with a layer of such microcapsules exhibit active electrocatalytic properties in OER during water electrolysis in the alkaline medium and are characterized by the overpotential value of 259 mV and Tafel slope of 64.8 mV/dec.

**Author Contributions:** Conceptualization, Review, and Editing, V.P.T.; Investigation, A.A.M.; Data Curation A.A.G.; Writing—Original Draft Preparation, V.P.T. and A.A.G.; Investigation of the electrochemical properties, E.V.B. All authors have read and agreed to the published version of the manuscript.

**Funding:** The study was supported by the RSF grant (project # 18-19-00370P). We are grateful to the "Nanotechnology" and "Physical Methods of Surface Investigation" Research Parks of Saint Petersburg State University, for their technical assistance in investigating the synthesized samples.

**Institutional Review Board Statement:** Not applicable.

**Informed Consent Statement:** Not applicable.

**Data Availability Statement:** Not applicable.

**Conflicts of Interest:** The authors declare no conflict of interest.

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
