# Peer review of "The Effect of the Open Vase-like Microcapsules Formation with NiFe Double-Hydroxide Walls during Hydrolysis of the Mixture NiSO4 and FeSO4 Salt Solution Microdroplets Deposited on the Alkaline Solution Surface"

_colloids, doi:10.3390/colloids6020032_

Round 1

Reviewer 1 Report

The manuscript under consideration deals with the to optimization of the conditions for synthesis of open vase-like microcapsules with Ni(II) and Fe(III) double hydroxide in view of the deposition of ordered arrays of such microcapsules on the nickel electrode surface. The paper is  organized in suitable sections with abstract and conclusions well describing aims and findings. Considering the worth of experimentals carried out in this work, the authors should provide more details about the Langmuir-Schaefer method applied to this case together with the surface features of the Ni foil.

Author Response

Thank you for these comments, we have taken them into account in the new version of the article. All text changes are marked in blue font.

Reviewer 2 Report

The manuscript presents the study of NiFe0.3(OH)x layered double hydroxide for oxygen evolution reaction (OER). The reviewer feels the manuscript needs to be improved before publication.

Major Points

  1. It has been well known that Pt counter electrode should not be used for water splitting.
  2. How would Ni/Fe ratio affect the OER performance? Why a 4:1 ratio was used for the feeding reagent?
  3. The source of Cu element in EDX should be explained (Figure 4A).
  4. It would be nice to have a comparison study of the OER performance of the samples prepared by LS and drop-casting methods.
  5. Long-term stability and the evolution of the structure and composition of the material should be presented.

Minor Points

  1. The title seems to be too long.
  2. Please try to improve the writing.

Author Response

Major Points

1. It has been well known that Pt counter electrode should not be used for water splitting.

Thanks for this remark, indeed, several articles are known in which it is shown that platinum electrodes are not recommended to be used as a reference electrode during electrolysis of water in an alkaline medium. In fact, we used a carbon rod as such an electrode. The fact that the platinum electrode is indicated in the first version of the article is a typo of our student who prepared the article for publication.

2. How would Ni/Fe ratio affect the OER performance? Why a 4:1 ratio was used for the feeding reagent?

This problem has been discussed in many articles and in the list of references we have given 2 new links to articles in which a similar ratio is used.

3. The source of Cu element in EDX should be explained (Figure 4A).

Please, see the explanation in the new version of the article.

4. It would be nice to have a comparison study of the OER performance of the samples prepared by LS and drop-casting methods.

Please, see the comparison of these properties in the new version of the article.

5. Long-term stability and the evolution of the structure and composition of the material should be presented.

 We have taken them into account in the new version of the article. All text changes are marked in blue font.

Minor Points

  1. The title seems to be too long.
  2. Please try to improve the writing.

These 2 comments are taken into account in the new version of the article. All changes in the text are highlighted in blue.

Round 2

Reviewer 2 Report

The revision has dealt with my previous comments.